# Human cytomegalovirus infection induces L1 expression through UL38-dependent mTOR-KAP1 pathway

Sehong Park[1,2,3], Jiseok Jeong[1,2,3], Kwangseog Ahn [1,2,3]*

1 Center for RNA Research, Institute for Basic Science, Seoul, Republic of Korea, 2 School of Biological Sciences, Seoul National University, Seoul, Republic of Korea, 3 SNU Institute for Virus Research, Seoul National University, Seoul, Republic of Korea

* ksahn@snu.ac.kr

## Abstract

Human cytomegalovirus (HCMV) and LINE-1 (L1) can co-inhabit a common host and closely interact with each other within a single cell. We have previously shown that HCMV exploits this opportunistic interaction by upregulating L1 expression that promotes its own productive life cycle by facilitating HCMV DNA replication. However, the mechanism by which HCMV increases L1 expression remains unknown. Here, we report that HCMV infection functionally inactivates KRAB-associated protein 1 (KAP1), a key epigenetic repressor of L1, through phosphorylation. HCMV infection of cells activates mTOR kinase that phosphorylates S824 residue of KAP1 and reduces its epigenetic repressive function, leading to increased chromatin accessibility of L1 promoter region. Treatment of potent mTOR inhibitor to the HCMV-infected cells was sufficient to reduce KAP1 phosphorylation and block L1 expression. Furthermore, cells infected with a mutant virus lacking UL38, an HCMV mTOR pathway activator, showed reduced KAP1 S824 phosphorylation and abolished L1 expression. Our results highlight the synergistic interaction between HCMV and L1 where HCMV UL38 serves as a primary viral regulator of L1 expression by upregulating the mTOR-KAP1 pathway.

## Introduction

Long-interspersed element-1 (LINE-1 or L1) is a retrotransposon that integrates into the host genome through reverse transcription, comprising approximately 17% of the human genome [1]. Structurally, L1 gene consists of 5' untranslated region (UTR), open reading frames 1 and 2 (ORF1 and ORF2), and 3'UTR. The L1 5' UTR contains an internal promoter activity, independent of upstream sequences [2]. L1 retrotransposition induces genomic instability in the host, a process associated with various diseases [3]. Under normal conditions, Krüppel-associated box (KRAB)-associated protein 1 (KAP1 or TRIM28) initiates L1 suppression by binding to the KRAB domain in the L1 region. KAP1 then recruits the human silencing hub (HUSH) complex, along with components of nucleosome remodeling and deacetylase (NuRD) complex via interaction with its SUMO domain, which maintains a transcriptionally repressive chromatin state through histone modification at the target locus [4–6]. Particularly,

**Data availability statement:** All relevant data are within the manuscript and its Supporting Information files.

**Funding:** This work was supported by the Institute for Basic Science of the Ministry of Science Grant (IBS-R008-D1) and the National Research Foundation of Korea (NRF) grant funded by the Korean government (NRF-2020R1A2C3011298) (to K. A.) and (NRF-2020R1A5A1018081) (to K.A.). The funders had no role in the study design, data collection, analysis, decision to publish, or preparation of the manuscript. This work was partially supported by funds from the SNU Institute for Virus Research.

**Competing interests:** NO authors have competing interests.

KAP1 has been shown to be a core regulator of L1 repression where its phosphorylation at S824 induces its de-SUMOylation and also dissociates other repressor components of HUSH and NuRD complex from the L1 loci, alleviating L1 repression [7, 8].

Human cytomegalovirus (HCMV), a member of the β-herpesvirus family, is a widespread pathogen that asymptomatically infects a significant portion of the global population [9]. HCMV has an approximately 230kb double-stranded DNA genome encoding numerous factors that modulate the host cellular environment, promoting the viral productive life cycle [10, 11]. Upon host cell infection, viral factors interact with epigenetic repressors, including KAP1, histone deacetylases, and death domain-associated protein 6, to regulate its latency and lytic cycle [12,13]. During lytic infection, HCMV utilizes viral factors to evade epigenetic silencing, which is necessary for establishing its productive life cycle [12–14]. However, whether these mechanisms also influence chromatin decondensation of L1 remains unknown.

HCMV infection is known to upregulate L1 expression [15,16]. However, the molecular mechanism by which L1 escapes epigenetic repression in response to HCMV infection remains unknown. L1 activation, contributing to genomic instability associated with various diseases [3], is induced by HCMV infection. Given that HCMV infects over 70% of the global population [9], it is crucial to understand how viral stimuli trigger L1 expression.

## Materials and methods

### Cells

Primary human foreskin fibroblast (HFF) and U373MG cells were cultured in Dulbecco's modified Eagle's medium (DMEM; HyClone) with 10% fetal bovine serum (FBS; HyClone), 1% (v/v) GlutaMAX-I (Gibco), and 100 U/mL Penicillin-Streptomycin (Gibco). Torin1 (Sigma-Aldrich; 1222998-36-8) was dissolved in DMSO and treated at 10 μM, 1 hour post-infection (hpi).

### Viruses

HCMV Toledo bacterial artificial chromosome (BAC) DNA was a gift from T. Shenk (Princeton University). Infectious HCMV particles were generated by transfecting primary HFFs with the BAC using the Neon Transfection System (Invitrogen). After a full cytopathic effect was observed, viral particles were harvested by centrifugation at 40,000g for 1 hour at 4°C. A UL38-deleted mutant of the Toledo BAC was engineered using the GeneBridge counter-selection BAC modification kit, replacing the UL38 region with a rpsL-neo cassette via homologous recombination. Primers for BAC recombination were designed as detailed in S1 Table. For HCMV titration, virus samples were serially diluted and used to infect HFFs for 1 hour. At 24 hpi, cells were fixed with 3.7% formaldehyde and permeabilized with 0.1% Triton X-100. Following incubation in 2% bovine serum albumin (BSA) in phosphate-buffered saline (PBS), cells were stained with a primary anti-HCMV IE1/2 antibody (Millipore; MAB810R), followed by a FITC-conjugated anti-mouse secondary antibody (Jackson Laboratories; 115-095-146). FITC-positive cells were counted to determine the number of infectious viral particles.

### Western blot

HFFs were infected with HCMV at 1 MOI. All samples were lysed using RIPA buffer (50 mM Tris pH 7.5, 150 mM NaCl, 1% Nonidet P-40, 0.5% sodium deoxycholate, 0.05% SDS, 1 mM EDTA) with 1 mM DTT, leupeptin, PMSF, protease inhibitor cocktail (ROCHE; 11836170001), and phosphatase inhibitor cocktail (Cell Signaling; 5872S) and boiled at 98 °C for 5 minutes. Protein concentrations were determined using the Pierce™ BCA Protein

Assay Kit (Thermo Fisher; 23225). Primary antibodies included anti-L1 ORF1p (Merck; MABC1152), anti-IE1/2 (Millipore; MAB810R), anti-UL44 (Virusys; CA006-1), anti-pp28 (Virusys; CA004-1), anti-KAP1/Phospho-KAP1 (Cell Signaling; ab10484/ab70369), and anti-S6K/Phospho-S6K (Cell Signaling; #9202/#9205). All primary antibodies were diluted 1:1000 in 5% BSA in Tris-Buffered Saline with Tween 20 (TBST), except for anti-L1 ORF1p (1:500). Peroxidase-conjugated secondary antibodies, anti-rabbit (Jackson; 111-035-003) and anti-mouse (Jackson; 115-035-062), were diluted 1:5000 in 5% skim milk in TBST. Detection was conducted using SuperSignal™ West substrate (Thermo Scientific; 34580, A38555). The experiment was repeated three times, and a representative image is shown.

The signal intensity for each experiment was measured using ImageJ software. The L1 ORF1p chemiluminescent intensity was normalized to the loading controls (Vinculin, HSC70), while the intensity of phospho-KAP1 and phospho-S6K was normalized to their respective total protein levels.

## Quantification of viral RNA and DNA via RT-PCR and qPCR

Total RNA was extracted using TRIzol reagent (Invitrogen; 15596026), treated with DNase I (Takara; 2270) at 37°C for 1 hour to remove genomic DNA, and then purified with TRIzol LS reagent (Ambion®; 10296–028). cDNA was synthesized using the ReverTra Ace qPCR RT Kit (Toyobo; FSQ-101). Genomic DNA was extracted using the QIAamp DNA Blood Mini Kit (QIAGEN; 51106) following the manufacturer's protocol. Real-time PCR was performed using the TOPreal qPCR 2X SYBR Green premix (Enzynomics; RT500M) with primers specified in the S2 Table.

## Chromatin accessibility assay

To assess the chromatin state at the endogenous L1 promoter region in HFFs infected with HCMV at 1 MOI, samples were collected at 72 hpi and analyzed using the Chromatin Accessibility Assay Kit (Abcam; ab185901). Chromatin state alterations were quantified by measuring Cq shifts after nuclease treatment and subsequent qPCR. Euchromatin regions, more sensitive to nuclease than heterochromatin, showed significant Cq changes. qPCR was conducted with primers targeting the L1 promoter and negative control regions (see S2 Table). Fold enrichment was calculated.

$$2^{[Cq(nuclease\ treated) - Cq(untreated)]}$$

## Luciferase assay

Cells were transfected with a luciferase reporter vector, infected with HCMV at 1 MOI after 24 hours, and analyzed for luminescence at each time point using the Dual-Luciferase Reporter Assay System (Promega; E1980). Each sample was treated with 200 μL of 1X passive lysis buffer, and 20 μL of lysate was used for luminometer measurements on the Microlumat PLUS LB 96V (Berthold). Firefly luminescence was normalized to Renilla luminescence to control transfection efficiency.

## RNA interference

siRNA targeting human KAP1 and non-targeting siRNAs (ON-TARGET plus, SMARTpool) were acquired from Dharmacon and introduced using DharmaFECT1 transfection reagent (Dharmacon). All siRNA was treated at 20 nM concentration.

## HCMV genome copy calculation

WT and mutant HFFs were infected with HCMV at an MOI of 1 and harvested at 24, 48, and 72 hpi. gDNA was extracted using the QIAamp DNA Blood Mini Kit and analyzed by qPCR. Total genomic DNA was quantified using MDM2 primers, and HCMV genome levels were determined using UL44 primers. The HCMV genome was normalized to MDM2 to ensure accurate quantification. To determine the HCMV copy number, a standard curve was generated using HCMV Toledo BAC DNA, serially diluted from 2 pg to 2 ng in HFF genomic DNA, maintaining a total DNA input of 20 ng. This standard curve was used to quantify viral genome copies in the samples.

## Statistical analysis

Results represent data from at least three independent experiments. Statistical analysis and graph generation were performed using GraphPad Prism 8.00 software. An unpaired Student's t-test was applied to compare the groups. A two-way analysis of variance (ANOVA) with Tukey's or Sidak's multiple comparisons test and one-way ANOVA with Tukey's multiple comparisons test were used for multiple group comparisons.

# Result

## HCMV infection upregulates the L1 expression by regulating L1 5'UTR promoter activity

We first analyzed the dynamics of L1 expression in primary human foreskin fibroblasts (HFFs) following 72 hours of HCMV infection. We found that L1 ORF1 protein expression increased throughout the 72-hour HCMV life cycle indicating that L1 expression was upregulated and L1 ORF1p accumulated over the course of HCMV infection (Fig 1A). To investigate whether the increase in L1 expression depended on *de novo* translated viral factors, we used UV-inactivated HCMV (UV-HCMV), which abrogates the expression of viral factors post-infection. HFFs infected with UV-HCMV did not induce L1 expression compared to the non-infection sample (Fig 1B). These results show that the expression of L1, which was scarcely expressed at the basal level, is significantly increased by the expression of HCMV factors after host cell entry.

With ATAC-sequencing, we previously found that the chromatin state of the endogenous L1 5'UTR promoter region decondensed after HCMV infection [15]. Based on this observation, we hypothesized that HCMV infection increases the L1 expression by regulating L1 promoter activity. To test this hypothesis, we utilized a dual-reporter vector containing firefly luciferase under the control of the L1 promoter and Renilla luciferase driven by the SV40 promoter that normalizes for reporter transfection efficiency (Fig 1C). 24 hours after reporter transfection, human glioblastoma cells (U373MG) were infected with HCMV at 1 MOI and reporter bioluminescence was measured at 24, 48, and 72 hours post-infection (hpi) (Fig 1C). Interestingly, HCMV infection progressively increased the firefly luminescence under L1 promoter throughout infection (Fig 1D). However, Renilla luciferase under SV40 promoter showed comparable expression between infected and non-infected samples at each time point, indicating that L1 promoter region is specifically activated during HCMV infection (Fig 1E). To further investigate whether the increase in promoter activity was due to increased chromatin accessibility of the L1 promoter region, we assessed the chromatin accessibility of L1 promoter region of HCMV-infected HFFs by measuring the difference in nuclease sensitivity of L1 promoter region. Both non-infected and HCMV-infected cells were treated with nuclease reaction mixtures, and the chromatin from each sample was extracted. We then quantitatively

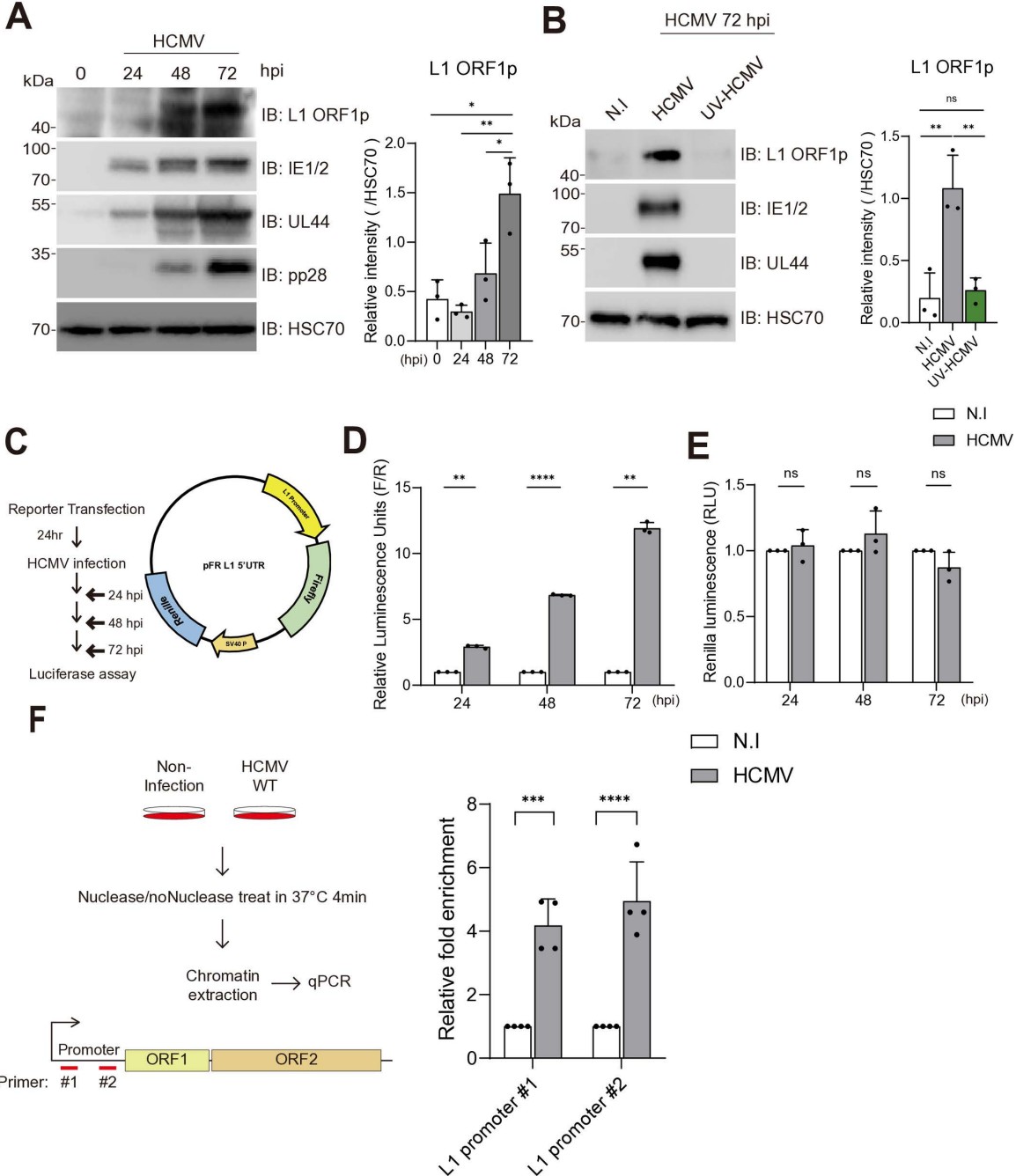

**Fig 1. HCMV infection upregulates L1 expression and L1 promoter activity through epigenetic modulation.** (A, B) Viral factors and L1 ORF1p expression in HFFs infected with HCMV at 1 MOI analyzed by Western blot. The chemiluminescent intensity of L1 ORF1p, measured using ImageJ, was normalized to the signal intensity of HSC70, which was used as a loading control. One-way ANOVA. (A) Time-course analysis of cells harvested at 24, 48, and 72 hpi. (B) Comparison of cells infected with active- or UV-HCMV harvested at 72 hpi. (C) Schematic representation of the luciferase assay (Left) and luciferase reporter vector (Right). (D, E) Relative bioluminescence in U373MG cells transfected with the luciferase reporter plasmid from (C) right. Data were normalized to the values of the non-infection sample. Two-way ANOVA. (D) Firefly luminescence normalized to Renilla luminescence to measure L1 promoter activity. (E) Renilla luminescence of HCMV-infected cells normalized to non-infected control. (F) Schematic of the chromatin accessibility assay with non-infected and HCMV-infected cells. Fold enrichment was calculated as $2^{([Cq \text{ (nuclease treated)} - Cq \text{ (untreated)}])}$ and each dataset was normalized to non-infection sample (Left). Chromatin accessibility of the endogenous L1 promoter region in HFFs, with L1-specific primers used for promoter detection (Right). Two-way ANOVA. Data is presented as mean ± standard deviation (SD). Statistical significance: ns (not significant) p > 0.1234, *p < 0.0332, **p < 0.0023, ***p < 0.0002, ****p < 0.0001.

measured the amount of extracted L1 promoter region by qPCR using two different primer sets targeting the L1 promoter region and calculated the shift in Cq values between nuclease-treated and non-treated samples (Fig 1F). Interestingly, HCMV-infected cells showed significantly higher chromatin accessibility of L1 promoter region compared to non-infected controls (Fig 1F). Collectively, these results indicate that HCMV infection increases L1 promoter activity by decondensing chromatin of this region.

## HCMV infection induces KAP1 phosphorylation and L1 derepression

Next, we investigated the mechanisms through which HCMV decondenses chromatin at the L1 promoter region. A number of epigenetic repressors have been reported to downregulate L1 expression where the functional loss of these repressors is sufficient to induce L1 expression [4, 5, 17]. We focused on KAP1, a key epigenetic regulator of L1 that recruits multiple epigenetic silencers to repress L1 expression [18]. To investigate whether KAP1 functions as the major epigenetic repressor of L1 in primary HFFs, we first transfected KAP1 siRNA and found that KAP1 was knocked down by 90% in siRNA-treated HFFs over control (Fig 2A and 2B). Expectedly, in siRNA-treated HFFs, L1 expression was increased by 49% compared to control samples, showing that KAP1 repression of L1 is also conserved in HFFs (Fig 2B). KAP1 has been shown to lose its repressive function when phosphorylated at residue S824 and the host cell controls KAP1 activity by optimal phosphorylation at this site [19]. We posited that HCMV infection induces L1 derepression by altering the activity of KAP1 via post-translational modification since the amount of KAP1 increased in HCMV-infected HFF (Fig 2C). Intriguingly, we found that the S824 phosphorylated KAP1 to KAP1 ratio progressively increased following HCMV infection (Fig 2C). UV-HCMV did not increase KAP1 phosphorylation, indicating that *de novo* translated viral factors produced post-infection inactivated KAP1, in accordance with our L1 induction data (Fig 2D). Taken together, these results demonstrate that upon HCMV infection, viral factors inactivate KAP1 by S824 phosphorylation, leading to an increase in L1 expression. Consequently, L1 ORF1p progressively accumulates in line with KAP1 phosphorylation.

## Activation of the mTOR by HCMV infection drives KAP1 S824 phosphorylation

The mTOR pathway is highly upregulated during HCMV infection and kinases within the signaling pathway are known to phosphorylate KAP1 [20]. Thus, we hypothesized that kinase activated by the mTOR pathway phosphorylated KAP1 upon HCMV infection. We first assessed the phosphorylation status of S6 kinase (S6K) T389, a downstream target of mTOR [21], as a marker of mTOR activation upon HCMV infection (Fig 3A). With T389 phosphorylation-specific antibody, we found that S6K T389 phosphorylation was significantly elevated upon HCMV infection, while no observable changes were detected in UV-HCMV-infected and non-infected control cells consistent with the previous report on HCMV infection-induced mTOR activation (Fig 3A) [22]. Next, we investigated whether HCMV-mediated mTOR activation phosphorylates KAP1. To this end, HCMV-infected cells were treated with Torin 1, a selective ATP-competitive inhibitor targeting mTOR kinase (Fig 3B). Torin 1 treatment for 72 hours at 10 μM concentration effectively inhibited mTOR observed by the loss of S6K T389 phosphorylation. Interestingly, Torin 1 treatment for HCMV-infected HFFs reduced both KAP1 phosphorylation and L1 expression in these cells (Fig 3C), indicating that mTOR pathway activation is crucial for KAP1-dependent L1 derepression. To further test the role of mTOR in the L1 chromatin state, we again assessed the L1 promoter chromatin accessibility after treating HCMV-infected HFFs with Torin 1. While HCMV infection

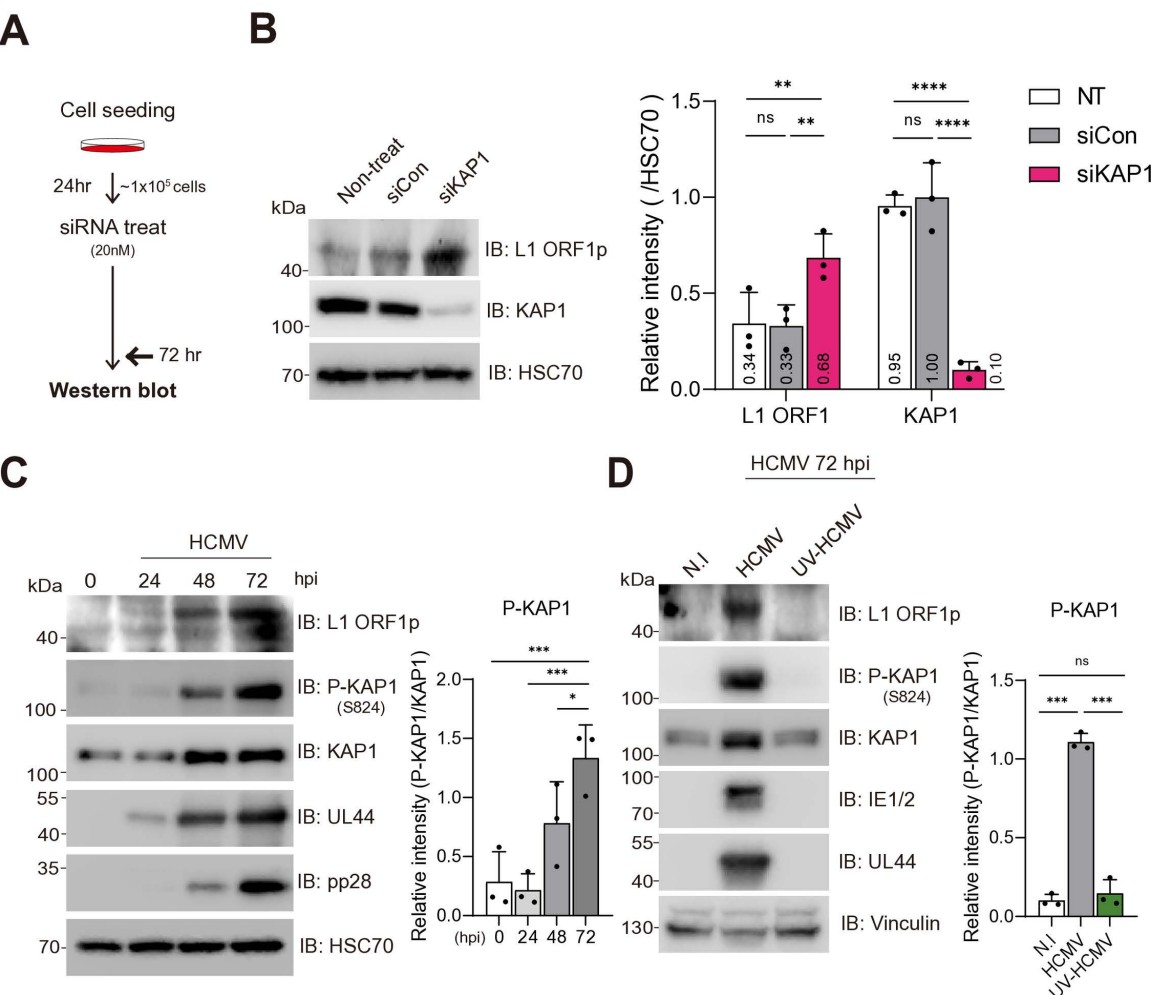

**Fig 2. HCMV activates the mTOR pathway, phosphorylating KAP1 through viral factors.** (A) A schematic representation of the siRNA treatment procedure. (B) Western blot analysis of KAP1 knockdown and L1 ORF1p expression levels in HFFs transfected with 20 nM siRNAs for 72 hours. The chemiluminescent intensities for KAP1 and L1 ORF1p were quantified with ImageJ software and were normalized to HSC70 signal intensity. One-way ANOVA. (C, D) Immunoblot analysis of KAP1 post-translational modifications during HCMV infection in HFFs. The signal intensity of P-KAP1, measured using ImageJ, was normalized to the signal intensity of total KAP1. One-way ANOVA. (C) Temporal analysis of KAP1 and phospho-KAP1 protein levels at 24, 48, and 72 hpi. (D) KAP1 phosphorylation was assessed in response to viral factors in HFFs infected with either active or UV-inactivated HCMV. Data is presented as mean ± standard deviation (SD). Statistical significance: ns (not significant) p > 0.1234, *p < 0.0332, **p < 0.0023, ***p < 0.0002, ****p < 0.0001.

increased the chromatin accessibility of the L1 promoter region, Torin 1 treatment completely abrogated the increase in the infected cells (Fig 3D). These results demonstrate that HCMV-activated mTOR kinases are necessary for KAP1 phosphorylation and subsequent epigenetic derepression of L1.

## HCMV UL38 mediated-mTOR activation relieves L1 repression during HCMV infection

HCMV UL38 is a known viral mTOR activator that antagonizes the mTOR inhibitory factor TSC2 [23]. To test if UL38 regulates L1 expression by upregulating the mTOR pathway, we generated UL38 deletion mutant virus (ΔUL38) with standard bacterial artificial chromosome recombineering [24] (Fig 4A). First, at 24, 48, and 72 hpi, UL38 RNA expression was detected

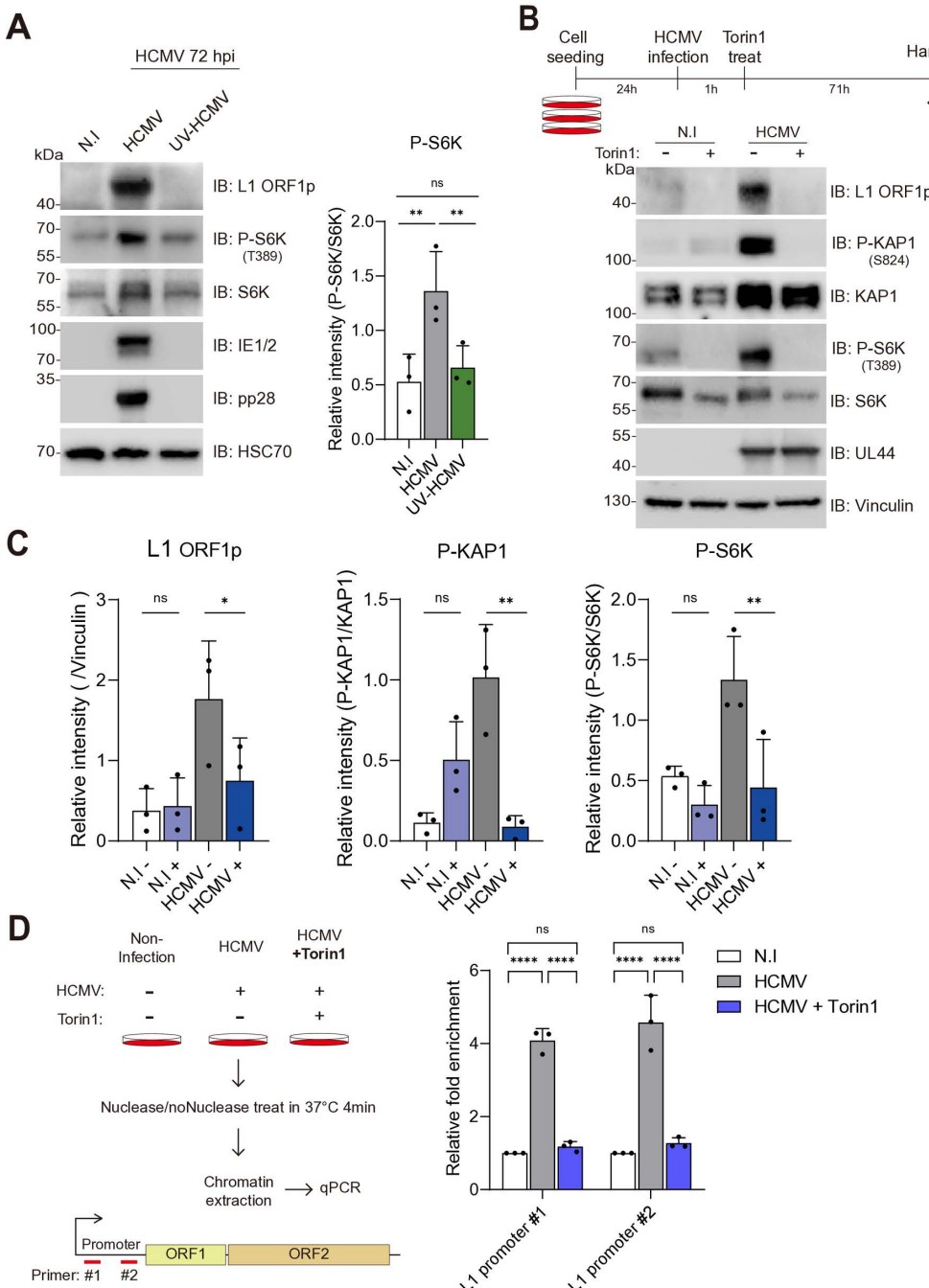

**Fig 3. Activation of the mTOR pathway by HCMV infection induces KAP1 inactivation.** (A) Western blot assay of HFFs infected with HCMV or UV-HCMV, assessing mTOR pathway activation. The chemiluminescent intensity of P-S6K, measured using ImageJ, was normalized to the signal intensity of total S6K. One-way ANOVA. (B, C) The effect of Torin1 treatment on L1 ORF1p expression and the phosphorylation levels of KAP1 and S6K were assessed via Western blot assay. HFFs were infected at 1 MOI and treated with 10 μM Torin1 at 1 hpi, and samples were harvested at 72 hpi. (C) L1 ORF1p signal intensity was normalized to Vinculin, while P-KAP1 and P-S6K signal intensities were normalized to total KAP1 and total S6K, respectively. One-way ANOVA. (D) Chromatin accessibility assay on HFFs infected with HCMV and treated with 10 μM Torin1 or DMSO (control), harvested at 72 hpi, assessing chromatin accessibility in the L1 promoter region. Data is presented as mean ± SD. Two-way ANOVA, ns p > 0.1234, *p < 0.0332, **p < 0.0023, ***p < 0.0002, ****p < 0.0001.

in wild type (WT)-infected cells but was absent in ΔUL38-infected cells (Fig 4B) Next, we assessed whether the deletion of UL38 affected viral genome entry compared to the WT at 3 hpi. The result showed that the deletion of UL38 did not impact the viral genome entry of the mutant virus into HFFs compared to the WT virus. (Fig 4C). In line with its known role in the mTOR pathway, ΔUL38-infected cells showed reduced S6K T389 phosphorylation (Fig 4D). ΔUL38-infected cells also showed reduced levels of phosphorylated KAP1 although the total amount of KAP1 protein was comparable with that of WT-infected HFFs. Importantly, ΔUL38-infected cells exhibited significantly reduced L1 ORF1p expression at all viral infection stages compared to WT-infected cells (Fig 4D). These results show that HCMV UL38 regulates L1 expression by inactivating KAP1 through the mTOR pathway. To further investigate whether UL38 contributes to increased L1 chromatin accessibility, we conducted chromatin accessibility assays in non-infected, WT-infected, and ΔUL38-infected HFFs in parallel at 72 hpi. While we observed a significant increase in chromatin accessibility of the L1 promoter region with the WT-infected sample, we did not observe any significant difference between ΔUL38-infected and non-infected samples (Fig 4E). Taken together, our data demonstrate that UL38-mediated mTOR activation inhibits chromatin repressor KAP1 via phosphorylation leading to increased chromatin accessibility of L1 promoter region and subsequent L1 expression (Fig 5).

## Discussion

Our study demonstrates how HCMV upregulates L1 expression primarily by phosphorylating KAP1 through the mTOR pathway. We identified HCMV-encoded mTOR activator UL38 as a necessary component for KAP1 inactivation and L1 derepression. Additionally, we found that HCMV UL38 contributes to the heterochromatin decondensation of the L1 promoter region through the mTOR-KAP1 pathway. Decondensation of L1 through this mechanism could facilitate the binding of L1 transcriptional activators, such as RUNX3 and YY1, whose expression we previously observed to increase upon HCMV infection, further promoting L1 expression [15].

Previous studies have shown that HCMV infection phosphorylates KAP1 via the mTOR pathway, as demonstrated using the mTOR inhibitor [13]. Nonetheless, the effect of HCMV infection-induced KAP1 phosphorylation on chromatin state has not been investigated. Our results show that HCMV-activated mTOR signaling increases chromatin accessibility at L1 promoter region through KAP1 post-translational modifications. Of note, KAP1 is known to function as a silencer not only for L1 but also for other retroelements such as SINE-VNTR-Alu and long terminal repeats [25]. Its depletion even without viral infection activates multiple major retrotransposon classes, highlighting the role of KAP1 as a global epigenetic repressor of retroelements [26]. Our findings extend this established knowledge by demonstrating that the UL38 induces epigenetic derepression by modulating cellular epigenetic regulator KAP1 and further implicates that this mechanism may have broader effects on multiple retroelements beyond L1, a largely unexplored area in the field. Additionally, KAP1 also inhibits the transcription of human immunodeficiency virus type 1 (HIV-1) [27]. Previous studies have demonstrated that HIV-1 is more active in HCMV-infected cells, which may be linked to the mTOR-KAP1 pathway demonstrated in our study [28, 29]. Similar to the mutualistic relation we have found between L1 and HCMV [15], infectious viruses may have other understudied symbiotic relations with host retroelements and viruses which are of particular interest for future investigations.

HCMV UL38 inhibits cellular apoptosis and activates the mTOR pathway by antagonizing TSC2, the cellular mTOR inhibitory factor [23, 30]. Our work reveals a novel role of UL38 in

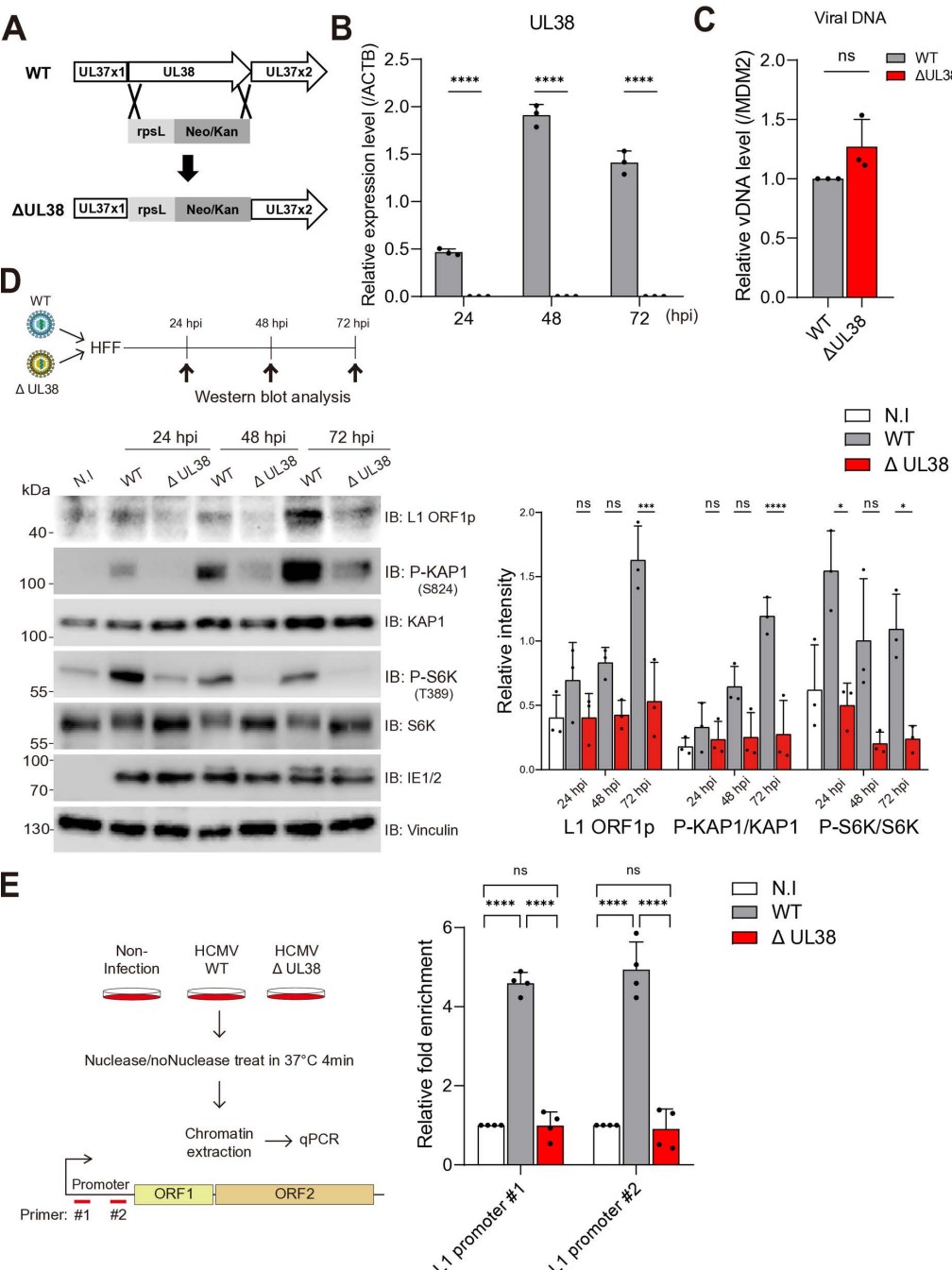

**Fig 4. HCMV UL38 enhances L1 expression by activating the mTOR pathway.** (A) Schematic illustrating the deletion of UL38 from HCMV Toledo using BAC modification. (B) Analysis of UL38 expression in RNA level. RNA harvested from samples at each time point was used to synthesize cDNA via RT-PCR. Two-way ANOVA. (C) Viral genome entry from samples at 3 hpi was analyzed. Viral DNA was detected using UL44 primers, while the host genome was detected using MDM2 primers. The amount of viral genome was normalized to the host genome to assess infectivity. Unpaired student's t-test. (D, E) Western blot and chromatin accessibility assays for HFFs infected with WT or ΔUL38 at an MOI of 1. (D) Temporal analysis of L1 expression and post-translational modifications of KAP1 and S6K by western blot assay. L1 ORF1p signal intensity was normalized to Vinculin, while P-KAP1 and P-S6K signal intensities were normalized to their respective total protein levels, total KAP1 and total S6K. Statistical significance was determined using one-way ANOVA. (E) Chromatin accessibility at 72 hpi using primers targeting the L1 5'UTR promoter. Two-way ANOVA. Data is presented as mean ± SD. Two-way ANOVA, ns p > 0.1234, *p < 0.0332, **p < 0.0023, ***p < 0.0002, ****p < 0.0001.

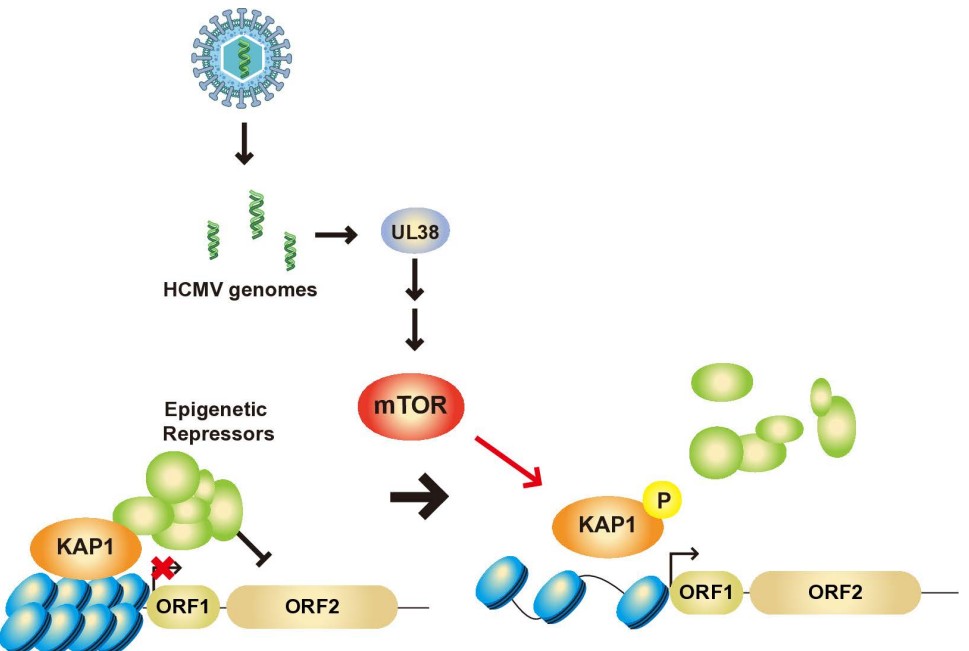

**Fig 5. Schematic model for activation of L1 expression by HCMV via the UL38-driven mTOR-KAP1 pathway.**
Upon entry, HCMV viral DNA undergoes de novo translation to produce UL38, which activates the mTOR pathway.
Activated mTOR kinase phosphorylates KAP1, leading to KAP1 functional inactivation and de-condensation of L1
heterochromatin to euchromatin, resulting in increased L1 expression.

L1 regulation, specifically highlighting its involvement in epigenetic regulation[15]. Furthermore, we have previously shown that L1 facilitates HCMV DNA replication via interactions between UL44 and L1 ORF2p [15]. Here, we showed that UL38 deletion reduces viral DNA replication, in line with previous findings where L1 knockdown in HCMV-infected cells produced similar effects (Fig S1B). Therefore, further investigation is required to clarify how UL38-mediated L1 regulation influences viral DNA replication, which may uncover a novel function of UL38.

HCMV establishes lifelong latency in the host and reactivates more frequently in older, immunocompromised individuals, contributing to various diseases [31–33]. Similarly, L1 activity increases with aging, and its expression promotes cellular senescence [34, 35]. The increased activation of both HCMV and L1 in the elderly population may be driven by their symbiotic interaction, as suggested by our findings [15]. Therefore, this research offers a novel perspective on the co-relationship between HCMV, L1, cellular senescence, and the diseases associated with aging.

In conclusion, our data highlights the novel factor of HCMV UL38 as the mTOR-KAP1 pathway regulator, leading to chromatin decondensation of the L1 retrotransposon. This finding enhances our understanding of how HCMV increases L1 expression through cellular mechanisms and suggests that UL38 may play a significant role in host-virus interactions by involving in epigenetic regulation.

## Supporting information

**S1 Table. BAC recombineering primer sets.**
(XLSX)

**S2 Table. qPCR primer sets .**
(XLSX)

**S1 File. Raw images.**
(PDF)

**S1 Fig. UL38 induces growth defects by affecting viral DNA replication.** (A) HFF cells were infected with WT and ΔUL38 virus at 0.1 MOI. Cell-free supernatants were harvested at each time point, diluted, and subjected to titration. The image at 9 dpi shows IE1/2 (green) and DAPI (blue). Two-way ANOVA. (B) HFFs were infected with the indicated HCMV at an MOI of 1 for each virus. Viral load was quantified by qPCR using (UL44/MDM2) as targets. Data is presented as mean ± SD from three independent samples (n = 3). Two-way ANOVA. ns p > 0.1234, *p < 0.0332, **p < 0.0023, ***p < 0.0002, ****p < 0.0001.
(TIF)

## Acknowledgment

We are thankful to Prof. Thomas Shenk (Princeton University) for providing HCMV Toledo. And we are also grateful to J.L. Garica-Perez, W.An, A. Roy-Engel, and A.J.Doucet for providing the L1 plasmids. We would like to express sincere gratitude to all members of Ahn laboratory for discussion and technical assistance.

## Author contributions

**Conceptualization:** Sehong Park.

**Data curation:** Sehong Park.

**Formal analysis:** Sehong Park, Jiseok Jeong.

**Investigation:** Sehong Park, Jiseok Jeong.

**Project administration:** Kwangseog Ahn.

**Supervision:** Kwangseog Ahn.

**Validation:** Sehong Park.

**Visualization:** Sehong Park, Jiseok Jeong.

**Writing – original draft:** Sehong Park.

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
