## [Editor Report · Decision Letter 0]

27 Nov 2024

PONE-D-24-53270Human cytomegalovirus infection induces L1 expression through UL38-dependent mTOR-KAP1 pathwayPLOS ONE

Dear Dr.  Ahn,

Thank you for submitting your manuscript to PLOS ONE. After careful consideration, we feel that it has merit but does not fully meet PLOS ONE’s publication criteria as it currently stands. Therefore, we invite you to submit a revised version of the manuscript that addresses the points raised during the review process.

We look forward to receiving your revised manuscript.

Kind regards,

Gianmarco Ferrara, PhD, MVD

Academic Editor

PLOS ONE

Journal Requirements:

4. PLOS ONE now requires that authors provide the original uncropped and unadjusted images underlying all blot or gel results reported in a submission’s figures or Supporting Information files. This policy and the journal’s other requirements for blot/gel reporting and figure preparation are described in detail at https://journals.plos.org/plosone/s/figures#loc-blot-and-gel-reporting-requirements and https://journals.plos.org/plosone/s/figures#loc-preparing-figures-from-image-files. When you submit your revised manuscript, please ensure that your figures adhere fully to these guidelines and provide the original underlying images for all blot or gel data reported in your submission. See the following link for instructions on providing the original image data: https://journals.plos.org/plosone/s/figures#loc-original-images-for-blots-and-gels. In your cover letter, please note whether your blot/gel image data are in Supporting Information or posted at a public data repository, provide the repository URL if relevant, and provide specific details as to which raw blot/gel images, if any, are not available. Email us at plosone@plos.org if you have any questions.

Additional Editor Comments:

Full-blot images are not included in the submission as supplementary files. We ask the authors to include these images in the format they deem appropriate (the molecular weight and the affiliation of the protein with the corresponding actin must be clear in the image). Thank you for your understanding. Before having these files, the paper cannot be revised.

---

## [Author Response · Author response to Decision Letter 1]

3 Dec 2024

Journal Requirements:

1.PLOS ONE style requirements:

The manuscript has been reformatted to fully adhere to the PLOS ONE style templates.

File names have been updated to comply with the journal's specified naming conventions.

2.Funding information:

All funding-related text has been removed from the manuscript, except for the details provided within the Funding Statement section in the online submission form, as per journal guidelines.

3.Supporting information captions:

Captions for all Supporting Information files have been included at the end of the revised manuscript. Corresponding in-text citations have been updated to maintain consistency throughout the document.

4.Original blot images:

Full, uncropped blot images underlying all results have been included as supplementary files.

These images are annotated to indicate molecular weights and alignment with appropriate controls.

All figures have been formatted to comply with the journal’s requirements for reproducibility and data presentation.

Additional Editor Comments:

All requested full-blot images are now included as supplementary files. These images include clear molecular weight markers and alignments with the corresponding actin controls, ensuring compliance with the journal's standards for figure data integrity and reproducibility.

Detailed annotations and descriptions of the raw blot images have been provided in the supplementary materials, as per the submission guidelines.

If any further adjustments or clarifications are required, please let us know, and we will address them promptly. Thank you for your valuable feedback and for considering our manuscript for publication.

---

## [Decision Letter · Decision Letter 1]

15 Jan 2025

PONE-D-24-53270R1Human cytomegalovirus infection induces L1 expression through UL38-dependent mTOR-KAP1 pathwayPLOS ONE

Dear Dr. Ahn,

Thank you for submitting your manuscript to PLOS ONE. After careful consideration, we feel that it has merit but does not fully meet PLOS ONE’s publication criteria as it currently stands. Therefore, we invite you to submit a revised version of the manuscript that addresses the points raised during the review process.

**ACADEMIC EDITOR: ** I ask the authors to respond and address the reviewer 4 comments

We look forward to receiving your revised manuscript.

Kind regards,

Gianmarco Ferrara, PhD, MVD

Academic Editor

PLOS ONE

Reviewers' comments:

Reviewer's Responses to Questions

**Comments to the Author**

1. If the authors have adequately addressed your comments raised in a previous round of review and you feel that this manuscript is now acceptable for publication, you may indicate that here to bypass the “Comments to the Author” section, enter your conflict of interest statement in the “Confidential to Editor” section, and submit your "Accept" recommendation.

Reviewer #1: (No Response)

Reviewer #2: (No Response)

Reviewer #3: (No Response)

Reviewer #4: (No Response)

2. Is the manuscript technically sound, and do the data support the conclusions?

Reviewer #1: Partly

Reviewer #2: Yes

Reviewer #3: Yes

Reviewer #4: Partly

3. Has the statistical analysis been performed appropriately and rigorously? 

Reviewer #1: I Don't Know

Reviewer #2: Yes

Reviewer #3: Yes

Reviewer #4: No

4. Have the authors made all data underlying the findings in their manuscript fully available?

Reviewer #1: Yes

Reviewer #2: Yes

Reviewer #3: Yes

Reviewer #4: Yes

5. Is the manuscript presented in an intelligible fashion and written in standard English?

Reviewer #1: Yes

Reviewer #2: Yes

Reviewer #3: Yes

Reviewer #4: Yes

6. Review Comments to the Author

Reviewer #1: I did not see their responses to the reviewers. I did not see their responses to the reviewers. I did not see their responses to the reviewers.

Reviewer #2: The research reveals that HCMV UL38 serves as a key viral regulator of L1 expression by activating the mTOR-KAP1 pathway. This interaction demonstrates a cooperative relationship between HCMV and L1, with HCMV leveraging L1 to boost its replication. Unraveling this mechanism sheds light on how HCMV manipulates host cellular functions and its possible effects on genomic stability and disease progression. However, the study primarily focused on the UL38 protein, potentially overlooking other viral or host factors involved in L1 activation. Additionally. the article would be enhanced by expanding the discussion to include the broader implications of the mTOR-KAP1 pathway, such as its involvement in other viral infections or diseases.

Reviewer #3: In this study Park and Ahn demonstrated that through the HCMV viral protein UL38 activates mTOR that phosphorylates KAP1, reducing its epigenetic repression and leading to increased chromatin accessibility of L1 promoter region. The study well conducted and well written and contributes to the understanding of the mechanism by which HCMV leads to expression of the LINE-1 retrotransposon. I believe that the manuscript has original data significant to the field and clarity of presentation and is suitable for publication in this current form.

Reviewer #4: The authors present compelling evidence that KAP1 phosphorylation downstream of mTOR signaling induces expression of the ORF1 protein of L1 retrotransposon, and that HCMV UL38 is required for this induction. While the data are convincing, most of the protein expression data are not quantitated and it is not clear if experiments were performed more than once, making it difficult to accept the conclusions drawn by the authors. Furthermore, the study could be strengthened by additional experiments to determine whether HCMV UL38 is sufficient to induce expression of L1 and whether these effects have implications for viral replication as has been previously reported for L1. Additionally, the authors should take care to more clearly explain how data was quantified as the results section and figure legends do not provide sufficient detail. Lastly, the authors at times draw conclusions that are not directly supported by the data and so should amend their statements to more accurately reflect their results. Major and minor concerns are listed below.

Major concerns:

1. Western Blots are not quantified, and it is not clear if each Western blot was performed more than once. The authors draw conclusions about changes (or lack of changes) in protein expression, but this should be quantified for the blots shown and a graph should be made from at least three replicate experiments. For an example, the authors state that total KAP1 levels “did not decrease” as shown in Fig 2B, but the levels appear to increase. This should be quantified and discussed in the revised manuscript. Quantification of blots needs to be done for Fig 1A-B, 2A-C, 3A-B, 4D.

2. Lines 150-152: “We found that L1 ORF1 protein expression increased throughout the 72-hour HCMV life cycle indicating that L1 expression was upregulated and L1 ORF1 accumulated over the course of HCMV infection (Fig 1A).” At what time points do you observe increased L1 expression? It does not appear to increase at 24hr, but this is not quantified. Is the upregulation at later time points indicative of when during the viral life cycle L1 is induced? Does this correlate with UL38 expression? Is it dependent on viral gene expression, DNA synthesis, etc? These points should be discussed in the manuscript.

3. Fig 1B. and 3A. Authors state that UV-HCMV does not induce L1 expression or phospho/total S6K, respectively, but there appears to be a slight induction. This should be quantified and discussed in the revised manuscript.

4. Your data is clear that UL38 expression is necessary for driving L1 expression, but is UL38 sufficient to increase mTOR signaling and induction of L1 expression? This could be tested using a plasmid expressing UL38. Or are other HCMV factors required to stimulate L1 expression?

5. Previous studies by this group showed that L1 expression during HCMV infection is important for viral replication. This would suggest that a UL38 mutant, which results in reduced L1 levels during infection, would exhibit a growth defect. The authors should provide growth curves for the UL38 mutant virus. Additionally, the authors should provide more discussion about how the findings in this study fit in with the data previously published for L1 expression during HCMV infection.

Minor concerns:

1. Fig 1D-E. Is the data RLU or is it relative to non-infected cells (set to 1)? Either is fine, but please describe this in more detail in the figure legend.

2. The figure legend for Fig 1F refers to up/down panels, but it should be left/right.

3. Also, for the Fig 1F legend please describe how the data is shown. It is unclear what “relative fold enrichment” means when the results state that the calculated shift in Cq values between nuclease-treated and non-treated samples was calculated.

4. Lines 56-58 and 334-335 are missing references.

5. One lines 205-208, the %knockdown and %increase in expression are not consistent with the data shown in Fig 2A.

6. Lines 216-218: “upon HCMV infection, viral factors inactivate...”: you have not thus far shown that it is a viral factor, just infection itself (but not likely entry since you don’t observe changes with UV-HCMV). This could be indirect effects. Please change the statement to something that more accurately reflects the data. Either HCMV infection induces, or HCMV viral gene expression induces… Your data suggests it is at later time points (48-72 hr), suggesting that it is not IE gene expression.

7. Fig 2A. legend does not fully explain values quantified. Were signal intensities quantified by ImageJ and normalized to a loading control?

8. On lines 267-269, it is stated that HCMV entry was unaffected. This is not exactly what the data show. The data shows equivalent levels of viral DNA, suggesting that there was no defect in entry, viral gene expression, etc. but the authors did not perform an entry assay. Please more accurately describe the results.

9. For Fig 4B-C, I cannot determine exactly what was measured based on the figure legend (what UL38 expression is normalized to, how viral DNA was measured). Also, the legend says that 4C shows UL38 expression when it does not. Please make these additions and changes.

10. In the discussion, there are multiple mentions of “the HCMV factor”. I’m assuming UL38, but this should be stated more clearly.

11. On lines 323-324, it is stated that, “the HCMV factor induces epigenetic derepression by interacting with cellular epigenetic regulator KAP1…” The data does not show a direct interaction, so this statement should be amended to reflect the data.

12. On lines 230 and 232, it should read, “the mTOR pathway” and not “mTOR pathway”. Also, on line 239, it should read “KAP1” and not “the KAP1”.

7. PLOS authors have the option to publish the peer review history of their article (what does this mean? ). If published, this will include your full peer review and any attached files.

**Do you want your identity to be public for this peer review?** For information about this choice, including consent withdrawal, please see our Privacy Policy .

Reviewer #1: No

Reviewer #2: No

Reviewer #3: No

Reviewer #4: No

---

## [Author Response · Author response to Decision Letter 2]

13 Feb 2025

February 13, 2025

Dear Dr. Gianmarco Ferrara,

Please find attached our revised manuscript, “Human cytomegalovirus infection induces L1 expression through the UL38-dependent mTOR-KAP1 pathway” (PONE-D-24-53270), for publication as a research article in PLOS ONE. The reviewers provided several important suggestions for revising our manuscript, which we have now fully addressed. In all cases where additional data was requested, we have provided new results, most of which have been incorporated into the manuscript. We have also addressed requests for clarification and additional explanation by modifying or adding new text. Lastly, the manuscript has been refined to align with the journal’s requirements. Altogether, these revisions have resulted in a more streamlined and polished study that provides new insights into the regulation of L1 expression by HCMV UL38.

We appreciate your consideration.

Sincerely,

Kwangseog Ahn

Professor

Seoul National University

Email: ksahn@snu.ac.kr

---

## [Decision Letter · Decision Letter 2]

20 Feb 2025

Human cytomegalovirus infection induces L1 expression through UL38-dependent mTOR-KAP1 pathway

PONE-D-24-53270R2

Dear Dr. Kwangseog Ahn,

We’re pleased to inform you that your manuscript has been judged scientifically suitable for publication and will be formally accepted for publication once it meets all outstanding technical requirements.

Kind regards,

Gianmarco Ferrara, PhD, MVD

Academic Editor

PLOS ONE

Additional Editor Comments (optional):

Reviewers' comments:

Reviewer's Responses to Questions

**Comments to the Author**

1. If the authors have adequately addressed your comments raised in a previous round of review and you feel that this manuscript is now acceptable for publication, you may indicate that here to bypass the “Comments to the Author” section, enter your conflict of interest statement in the “Confidential to Editor” section, and submit your "Accept" recommendation.

Reviewer #3: All comments have been addressed

Reviewer #4: All comments have been addressed

2. Is the manuscript technically sound, and do the data support the conclusions?

Reviewer #3: Yes

Reviewer #4: Yes

3. Has the statistical analysis been performed appropriately and rigorously? 

Reviewer #3: Yes

Reviewer #4: Yes

4. Have the authors made all data underlying the findings in their manuscript fully available?

Reviewer #3: No

Reviewer #4: Yes

5. Is the manuscript presented in an intelligible fashion and written in standard English?

Reviewer #3: Yes

Reviewer #4: Yes

6. Review Comments to the Author

Reviewer #3: (No Response)

Reviewer #4: The authors responded to all of my comments and also provided additional data to address my concerns.

7. PLOS authors have the option to publish the peer review history of their article (what does this mean? ). If published, this will include your full peer review and any attached files.

**Do you want your identity to be public for this peer review?** For information about this choice, including consent withdrawal, please see our Privacy Policy .

Reviewer #3: No

Reviewer #4: No

---

## [Editor Report · Acceptance letter]

PONE-D-24-53270R2

PLOS ONE

Dear Dr. Ahn,

I'm pleased to inform you that your manuscript has been deemed suitable for publication in PLOS ONE. Congratulations! Your manuscript is now being handed over to our production team.

Kind regards,

on behalf of

Prof. Gianmarco Ferrara

Academic Editor

PLOS ONE